# The Effect of Cataract Surgery on the Risk of Dementia: A Nationwide Cohort Study

**DOI:** 10.3390/jcm12206441

**Published:** 2023-10-10

**Authors:** Chaeyeon Lee, Eunhae Shin, Mina Kim, Yoonjong Bae, Tae-Young Chung, Sang Won Seo, Hyemin Jang, Dong Hui Lim

**Affiliations:** 1Department of Ophthalmology, Samsung Medical Center, Sungkyunkwan University School of Medicine, Seoul 06351, Republic of Korea; dudalsqk92@gmail.com (C.L.); taeyoung15.chung@samsung.com (T.-Y.C.); 2Seoul Nunevit Eye Clinic, Seoul 05551, Republic of Korea; getthedear@naver.com; 3Department of Data Science, Hanmi Pharmaceutical Co., Ltd., Seoul 05545, Republic of Korea; mina.kim92@hanmi.co.kr (M.K.); yoonjong.bae@hanmi.co.kr (Y.B.); 4Renew Seoul Eye Center, Seoul 06615, Republic of Korea; 5Alzheimer’s Disease Convergence Research Center, Samsung Medical Center, Seoul 06351, Republic of Korea; sw72.seo@samsung.com; 6Department of Neurology, Samsung Medical Center, Sungkyunkwan University School of Medicine, Seoul 06351, Republic of Korea; 7Department of Digital Health, SAIHST, Sungkyunkwan University, Seoul 06355, Republic of Korea; 8Department of Neurology, Seoul National University, Seoul 03080, Republic of Korea; 9Samsung Advanced Institute for Health Science & Technology, Sungkyunkwan University School of Medicine, Seoul 06351, Republic of Korea

**Keywords:** cataract, cataract surgery, dementia

## Abstract

Background: An advanced age and the female sex are widely recognized risk factors for both cataract and dementia. We investigated the effect of cataract surgery on the incidence of dementia in a Korean population aged ≥ 45 years with a previous diagnosis of cataract. Methods: This nationwide cohort study was performed using Korean National Health Insurance Service data collected from 2006 to 2017. A total of 300,327 subjects aged ≥ 45 years with a history of cataract diagnosis but no previous diagnosis of dementia were analyzed. The relationship between cataract surgery and dementia was evaluated, applying a time-varying analysis to evaluate the hazard ratio (HR) and 95% confidence interval (CI) values according to dementia. It was calculated via a multivariable Cox regression model, with adjustments for age, sex, visual acuity (VA), ocular and systemic comorbidities, and social factors (including body mass index, income, smoking, and drinking). Results: In the multivariate analysis, the cataract surgery group showed a marginal difference in dementia development (HR 1.10 [95% CI 1.02–1.19]) because both cataract and dementia share common risk factors. However, in the subgroup analysis, men (HR 0.49 [95% CI 0.26–0.90]) and patients under 65 years of age (HR 0.88 [95% CI 0.79–0.99]) in the group with cataract surgery and good VA showed a significantly lower incidence of dementia. Conclusion: Through visual improvement, together with timely surgical intervention, the procedure can alleviate the risk of dementia in visually impaired patients, especially in younger and male patients.

## 1. Introduction

Dementia is one of the most important age-related disorders characterized by progressive cognitive decline and a diminished capacity to carry out daily tasks [1,2]. The number of patients with dementia is increasing globally, which causes substantial health and economic burdens [2]. Cataract stands as the primary global cause of blindness, impacting 95 million individuals around the world [3]. Previous studies have discussed the relationship between cataract and dementia [4,5,6]. In a recent meta-analysis, it was found that cataract itself could increase the risk of dementia [5]. The analysis revealed a pooled hazard ratio (HR) of dementia of 1.17 (95% confidence interval (CI): 1.00–1.38) in cataract patients [5]. This relationship might be due to their sharing of common risk factors, particularly old age [4,5], hypertension (HTN) [4,5], diabetes mellitus (DM) [1,5], and a low education level [4]. Alternatively, this connection has also been explained by the presence of decreased visual acuity (VA) in cataract patients, which may affect cognitive impairment [4,7,8].

Considering that decreased VA attributed to cataract is associated with dementia risk, we could hypothesize that cataract surgery performed to improve VA may lower the risk of dementia. Many earlier studies have reported that there is no association between cataract surgery and the risk of dementia, while recently, in-depth studies showed conflicting results [4,6,9,10]. According to a meta-analysis exploring the impact of cataract surgery on cognitive function, greater cognitive improvement was expected after cataract surgery [9]. However, the sample size of all included articles was <400, and the follow-up period was <1 year [9]. Lee et al. argued that cataract surgery significantly lowered the development of dementia (HR 0.71 [95% CI, 0.62–0.83]; *p* < 0.001) [6]. Nonetheless, the total participant number was 3038, and the study investigators did not consider ophthalmic diseases or VA as compounding factors [6].

Therefore, in the present study, we assessed how cataract surgery influenced the occurrence of dementia in a nationwide, population-based cohort population. We used multivariate analysis by adjusting for various compounding factors, particularly VA and other ocular diseases, that may have affected the outcomes. We hypothesize that cataract surgery could lower the risk of dementia development when it comes to VA improvement.

## 2. Method and Material

### 2.1. Study Setting and Source of Data

The Korean National Health Insurance Service (NHIS) functions as the sole insurer in South Korea, offering mandatory universal medical insurance coverage to 97% of the Korean population. The government finances the Medical Aid program, which is extended to the remaining 3% of the population in the lowest income category. The NHIS collects clinical records of diagnosis and prescriptions for each patient and provides national health screening for all Koreans aged 40 and above. These collections include general health examinations and the recording of lifestyle behavior information through questionnaires.

The NHIS databank consists of collections of patient information encompassing eligibility details (such as age, gender, eligibility type, location, and income), claims data (which include general information, diagnosis based on the 10th revision of the International Classification of Disease [ICD-10], prescription records, and consultation records), health assessments (including self-reported questionnaires on health-related behaviors such as smoking, drinking, and medical history), laboratory tests (such as lipid and fasting glucose), physical measurements (such as blood pressure and body mass index (BMI)), and mortality [11,12]. 

This study was a retrospective cohort investigation conducted on a national scale, utilizing approved data sourced from the Korean NHIS database. The research adhered to the principals outlined in the Declaration of Helsinki and received approval from the Institutional Review Board (IRB) of the Samsung Medical Center (IRB file no. SMC 2020-04-146). The review board exempted the need for informed consent because the study data are publicly available and anonymized, as the study itself was retrospective.

### 2.2. Study Participants

A cohort without pre-existing dementia was constructed through random sampling from the NHIS data. Patients ≥ 45 years of age in 2006 and those not diagnosed with dementia between 2002 and 2005 were included. Among 799,681 individuals, those who were diagnosed with cataract or those who underwent cataract surgery between 1 January 2002 and 31 December 2016 were included (*n* = 304,726). Individuals who developed dementia before cataract diagnosis or surgery or those who developed dementia within 1 year from the diagnosis or surgery date were excluded (*n* = 4362). Those who had been disqualified from the NHIS before cataract diagnosis or surgery (*n* = 18) and individuals with missing data or errors in mortality data (*n* = 19) were excluded. Finally, the analysis encompassed total of 300,327 participants (Figure 1). Participants who were diagnosed with cataract were segregated into two groups: one undergoing cataract surgery and the other without cataract surgery. 

### 2.3. Variables for Exposure and Outcome

The exposure variable was cataract surgery, defined by both of the surgery codes S5119 (phacoemulsification) and S5117 (primary intraocular lens implantation) with ICD-10 codes for cataract (H25 and H26). 

The outcome variable was newly diagnosed dementia, identified by an ICD-10 code co-existing with ≥1 dementia drug prescription. Dementia drugs included N-methyl-D-aspartate receptor antagonist (memantine) or acetylcholinesterase inhibitors (rivastigmine, donepezil hydrochloride, and galantamine). The ICD-10 codes pertaining to dementia included the following: F00, F01, F02, F03, F05, G30, and G31. 

### 2.4. Other Variables and Follow-Up

Covariates were established as demographic variables (age, gender, and income), VA, lifestyle factors (BMI, smoking history, and alcohol consumption), and comorbidities. Comorbidities encompassed HTN, DM, dyslipidemia, stroke (identified via ICD-10 codes I63, I64, I65, and I66), depression (F32 and F33), chronic heart disease (CHD, I20, I21, I22, I23, I24, and I25), glaucoma (if the individuals had glaucoma surgery), diabetic retinopathy (DMR, E10.3, E11.3, E12.3, E13.3, E14.3, or H36.0), and age-related macular degeneration (AMD and H35.3). HTN was determined via ICD-10 codes (I10, I11, I12, I13, and I15) with prescription for HTN medication or via a systolic BP (SBP) ≥ 140 mmHg or diastolic BP (DBP) ≥ 90 mmHg revealed by means of a health screening test. DM was identified via ICD-10 codes (E10, E11, E12, E13, and E14) coupled with either a fasting blood glucose level ≥ 126 mg/dL detected during a health screening test or a prescription for DM medication. Dyslipidemia was defined with ICD-10 codes (E78 and I70) along with a prescription for dyslipidemia medication or a triglyceride level ≥ 240 mg/dL. 

Subjects were categorized into groups based on their VA, with one group having good VA (Snellen chart: ≥ 20/66) and the other group having poor VA (<20/66) based on the health screening tests. BMI was calculated by dividing an individual’s weight in kilograms by the square of their height in meters (kg/m^2^) and was further categorized into (1) underweight, <18.5 kg/m^2^; (2) normal, 18.5 to <23 kg/m^2^); (3) overweight, 23 to <25 kg/m^2^; and (4) obese, 25–30 kg/m^2^ [13]. Household income level was determined according to the national health insurance premium level, which was divided into 20 levels. Participants were grouped into four categories based on their premium level: (1) low, levels 1–5; (2) lower-middle, levels 6–10; (3) higher-middle, levels 11–15; and (4) high, levels 16–20. The study follow-up period lasted from 1 January 2006 to 31 December 2017, or until the date of dementia diagnosis.

Participants answered the questionnaire about how much alcohol they consumed in a week. By multiplying the frequency of drinking and the amount (g) per drinking, alcohol consumption was calculated. “Heavy drinker” was identified when the participant consumed more than 30 g of alcohol per day. Smoking behavior was divided into three categories: “never”, “ex”, and “current”. “Ex-smoker” was classified when the participant answered “no” to the question about current smoking but “yes” to the following question: “Have you smoked at least 100 cigarettes, 5 packs, during your lifetime?”

### 2.5. Statistical Procedures

We performed Kolmogorov–Smirnov test on continuous variables, and *p*-value were >0.05, which was consistent with normal distribution. Therefore, continuous variables were expressed as mean ± standard deviation (SD). Categorical variables were expressed as counts and percentage. Student’s *t* test and the chi-squared test were employed to assess the statistical differences among continuous and categorical variables. Univariate Cox proportional hazards regression analysis was utilized to compute the crude hazard ratio (HR) and 95% confidence interval (CI) values for dementia. Additionally, after adjustment for demographic factors (age, sex, and income), VA, behavioral factors (BMI, smoking history, and alcohol consumption), and comorbid conditions (HTN, DM, dyslipidemia, stroke, depression, CHD, glaucoma, DMR, and AMD), multivariate Cox proportional hazards regression analysis was conducted. A significance level of *p* < 0.05 was regarded as statistically significant. Statistical analyses were performed using the R statistical software program (Version 3.5; Foundation for Statistical Computing, Vienna, Austria) and SAS (Version 9.3; SAS Institute Inc., Cary, NC, USA).

## 3. Results

### 3.1. Initial Characteristics of the Study Cohort

Table 1 displays the initial characteristics of the individuals included in the study. The age and follow-up period in the surgery and non-surgery groups were normally distributed (Kolmogorov–Smirnov test, age, *p* = 0.2; follow-up period, *p* = 0.38). The subjects in the surgery group had a shorter mean follow-up period (6.08 ± 3.06 vs. 6.29 ± 3.75 years) and were older (67.79 ± 8.74 vs. 66.21 ± 8.94 years) than the subjects in the non-surgery group. There was a greater percentage of female patients in the surgery group (61.6% vs. 59.6%), and the proportions of subjects with DM (25.6% vs. 23.5%), HTN (66.4% vs. 63.1%), glaucoma (2.6% vs. 1.9%), DMR (15.2% vs. 9.2%), and AMD (17.6% vs. 7.8%), subjects who were never smokers (54.4% vs. 51.8%), and current smokers (9.3% vs. 9.1%) were greater in the surgery group. Conversely, the proportions of subjects with dyslipidemia (35.4% vs. 38.0%), depression (17.7% vs. 19.6%), and who practiced heavy drinking (9.5% vs. 10.4%) were lower in the surgery group.

### 3.2. Univariate Analysis

In the univariate analysis (Appendix A), cataract surgery was linked to an increased dementia development (HR 1.35 [95% CI, 1.32–1.38]). Age ≥ 65 (HR 8.28 [95% CI, 7.96–8.61]), female sex (HR 1.65 [95% CI, 1.61–1.70]), systemic (DM, HTN, stroke, depression, and CHD) and ocular comorbidities (DMR and AMD), and poor VA in the better eye were associated with an elevated dementia risk (*p* < 0.0001).

### 3.3. Multivariate Analysis

In the multivariate analysis, the cataract surgery group showed a marginally significant difference in dementia risk (model 4, HR [95% CI]: 1.10 [1.02–1.19]) after adjusting for the covariates (Table 2). In the subgroup analysis, the study population was categorized as the whole population, participants aged ≥ 65 and <65 years, male, and female, respectively (Table 3). In the <65 years old group, those with cataract surgery and a good VA exhibited a significantly reduced dementia risk (HR 0.88 [95% CI, 0.79–0.99]) compared to the individuals without surgery and a poor VA. In the male group, those with cataract surgery and good VA exhibited a reduced dementia risk (HR 0.49 [95% CI, 0.26–0.90]) compared to individuals without surgery and a poor VA. Also, those without surgery but a good VA had a reduced dementia risk (HR 0.49 [95% CI, 0.26–0.89]). No other subgroups had statistically significant results.

## 4. Discussion

In our nationwide population-based cohort study, we examined the effect of cataract surgery on the likelihood of developing dementia. Because both dementia and cataract share common risk factors and are predominant among aged women, exploring the influence of cataract surgery on dementia development is complicated. Our major study findings are as follows: First, in the multivariate analysis, the cataract surgery group had a slightly increased dementia risk (model 4, HR [95% CI]: 1.10 [1.02–1.19]) after adjusting for multiple covariates. Second, in the subgroup analysis, the cataract surgery with a good VA group showed a significantly reduced dementia risk compared to patients in the no surgery and poor VA group, especially when they were male and/or aged < 65 years old. 

In early studies, many investigators reported that cataract surgery did not relate to the risk of dementia, while more recently, in-depth studies have offered conflicting results [6,10,14,15]. In a study published in 2005, the enhancements in cognitive and visual abilities were not associated with cataract surgery, leading the authors to propose that cataract surgery has no impact on cognitive function [15]. However, this study was a single-center study with a relatively small number of 301 participants, and each group was not randomized [15]. Furthermore, since there was an improvement in cognitive function even in the group that was diagnosed with cataract but declined surgery, it is difficult to explain the conclusion [15]. On the contrary, according to a cross-sectional epidemiological analysis involving 2764 Japanese individuals published in 2018, the cataract surgery group had reduced risk for mild cognitive impairment compared to the no surgery group (odds ratio: 0.78 [95% CI: 0.64–0.96]; *p* = 0.02) [14]. However, there wase no significant difference observed in the risk for dementia (odds ratio: 1.10 [95% CI: 0.75–1.61]; *p* = 0.64) [14]. They explained that the reason for this difference is that MCI is a preventable and reversible condition [14].

Furthermore, in a retrospective study utilizing data from the Taiwan National Health Insurance Database, which included 113,123 participants aged 70 years or older who were diagnosed with cataract, the researchers determined that the cataract surgery group had a 0.74 times lower dementia risk (HR, 0.74 [95% CI: 0.75–0.79] *p* < 0.001) [10]. The preventive impact of cataract surgery on dementia was notably linked to being female (*p* < 0.001), while it showed no association based on the age groups (*p* = 0.413), which is different from our results [10]. As the follow-up period was shorter than ours (10 years) and dementia was diagnosed only with the diagnostic code, our analysis was more rigorous [10]. In a prospective cohort study with 3038 participants, patients who underwent cataract extraction showed a reduced risk (HR, 0.71 [95% CI, 0.62–0.83]; *p* < 0.001) of developing dementia compared to those who had not undergone such a surgical procedure [6]. However, those studies usually enrolled only a limited number of participants, and few conducted a multivariate analysis [6,10]. Moreover, previous articles neither adjusted for the effect of other ocular diseases that could decrease the VA nor adjusted for the effect of VA itself [6,10]. When we looked at this association, we considered not only general diseases (DM, HTN, dyslipidemia, stroke, depression, and CHD), but also ophthalmic diseases such as glaucoma [16], DMR [17], and AMD [18,19] since they are risk factors for both dementia and cognitive impairment. In the future, it is necessary to reveal whether cataract surgery is effective in dementia prevention through large-scale research in diverse populations.

Furthermore, VA itself is considered an independent risk factor for dementia [7,8]. Many studies have argued that visual acuity impairment is associated with an increased dementia risk [7,8]. Therefore, we adjusted for VA using data from the NHIS health screening test. VA is a potent risk factor in this study, as inferred from the univariate analysis result (Appendix A), suggesting that a poor VA is correlated with a 2.263-fold greater dementia risk. However, in the subgroup analysis, after adjusting for covariates (Table 3), individuals without cataract surgery and with a good VA had no significant difference in the dementia risk compared to the patients with a poor VA (*p* = 0.236); however, a significant difference was found among men (*p* = 0.0191). Among the whole population, across all ages and the male group, except for women, a good VA showed a lower risk for dementia, although the differences were not significant. Further research that will consider VA when exploring the relationship between cataract surgery and the dementia risk is necessary. 

As age and the female sex were also discussed as pivotal risk factors for dementia development based on previous studies [1,2], the impacts of risk factors such as old age (≥65 years), female sex, and a poor VA are enormous. Although the HR value of cataract surgery in patients with a good VA was lower than the reference value of 1 in the whole population and the aged ≥ 65 years old subgroup, it showed no significant difference. 

Nevertheless, both cataract surgery and a good VA lowered the risk of dementia in men and those aged < 65 years. Modern cataract surgery has a high efficacy with fewer safety concerns in older patients. Generally, cataract surgery operated by an experienced surgeon improves the VA with a low complication rate [20]. However, an increasing age is associated with worse VA outcomes following cataract surgery. With each additional year of age, the patients had a 0.96-fold lower likelihood of attaining a VA of ≥6/9 on the Snellen chart [21]. Although our study did not compare the VA before and after surgery, our study suggests that cataract surgery should be recommended at a proper time.

Cataract surgery involves various types of intraocular lenses (IOLs), such as monofocal, multifocal, or toric lenses [22]. These variations in the surgical approach and IOL choice are primarily aimed at addressing specific visual needs and correcting refractive errors [22]. However, the relationship between the type of cataract surgery and dementia risk is not a well-established factor in the decision-making process for cataract surgery [22,23]. According to authors who involved 25 patients with multifocal surgery and 26 patients with monofocal surgery, the IOL type did not show a significant difference in cognitive dysfunction [23]. However, the number of enrolled patients was small, and the covariates were not adjusted [23]. Further research considering this relationship is required with a large number of subjects.

Our study has several limitations. To begin with, being a retrospective study, the nature of this kind of study carries a limitation. In particular, the definition of each factor (disease definition and health status) might lead to an underestimation. Second, the causal relationship is difficult to reveal because the VA before and after surgery was incomparable. Nevertheless, this study contains a vast cohort sample that advocates for the validity of the statistical result. Furthermore, we tried to adjust for as many factors as possible to minimize the compounding effect on the dementia risk. Finally, the Korean population is mostly homogeneous in terms of ethnicity (Northeast Asian). Therefore, further studies with other races are required to extend these results. 

In our study, cataract surgery in patients with a good VA significantly lowered the risk of dementia in male patients or those aged < 65 years. Cataract surgery should be recommended at an appropriate time, especially in patients with potentially good vision after surgery, such as those with a younger age and a relatively good VA, as it can alleviate the risk of dementia. 

## Figures and Tables

**Figure 1 jcm-12-06441-f001:**
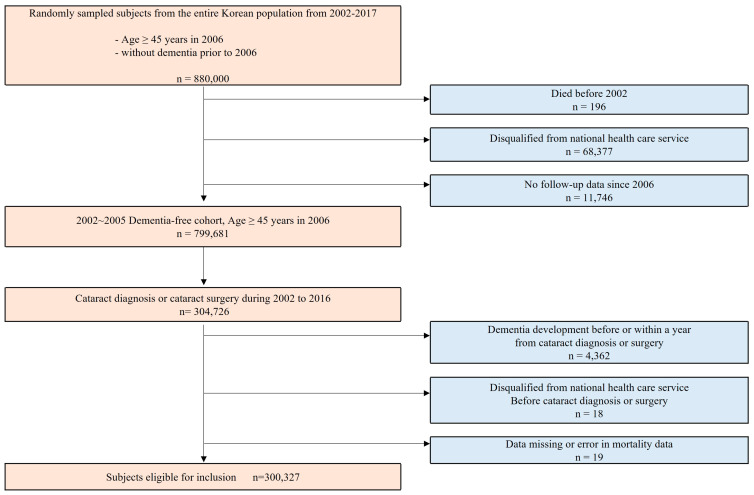
Flowchart of eligible subjects in the dementia-free cohort.

**Table 1 jcm-12-06441-t001:** The initial characteristics and the composition of the study cohort.

Variables	Total (*n* = 300,327)	Cataract Surgery		*p*-Value
		Yes (*n* = 111,208)	No (*n* = 189,119)	
*1. Demographic Factors*				
Age (mean ± SD)	66.79 ± 8.90	67.79 ± 8.74	66.21 ± 8.94	**<0.0001**
Sex, female (%)	181,253	68,489 (61.6%)	112,764 (59.6%)	**<0.0001**
Income (grade)				
Q1 (low)	72,395	26,909 (24.2%)	45,486 (24.1%)	0.5233
Q2 (lower-middle)	47,061	17,514 (15.7%)	29,547 (15.6%)	
Q3 (higher-middle)	68,595	25,301 (22.8%)	43,294 (22.9%)	
Q4 (high)	112,276	41,484 (37.3%)	70,792 (37.4%)	
Follow-up period (years)	6.83 ± 3.68	6.08 ± 3.06	6.29 ± 3.75	**<0.0001**
*2. Systemic Comorbidities*				
Dementia	32,895	14,016 (12.6%)	18,879 (10.0%)	**<0.0001**
DM	72,867	28,471 (25.6%)	44,396 (23.5%)	**<0.0001**
HTN	193,044	73,803 (66.4%)	119,241 (63.1%)	**<0.0001**
Dyslipidemia	111,242	39,325 (35.4%)	71,917 (38.0%)	**<0.0001**
Stroke	35,416	13,220 (11.9%)	22,196 (11.7%)	0.2150
Depression	56,792	19,636 (17.7%)	37,156 (19.6%)	**<0.0001**
CHD	80,856	29,776 (26.8%)	51,080 (27.0%)	0.1620
Glaucoma	6382	2840 (2.6%)	3542 (1.9%)	**<0.0001**
Diabetic retinopathy	34,361	16,868 (15.2%)	17,493 (9.2%)	**<0.0001**
AMD	34,234	19,528 (17.6%)	14,706 (7.8%)	**<0.0001**
*3. Behavioral Factors*				
BMI (kg/m^2^) ^a^				
<18.5	5773	2436 (2.2%)	3337 (1.8%)	0.6989
18.5 to <23	72,275	27,214 (24.5%)	45,061 (23.8%)	
23 to <25	58,184	21,676 (19.5%)	36,508 (19.3%)	
≥25	72,482	27,665 (24.9%)	44,817 (23.7%)	
Smoking ^b^				
Never smoked	158,460	60,525 (54.4%)	97,935 (51.8%)	**<0.0001**
Ex-smoker	28,190	10,417 (9.4%)	17,773 (9.4%)	
Current smoker	27,622	10,331 (9.3%)	17,291 (9.1%)	
Drinking (heavy)				
Heavy	30,235	10,576 (9.5%)	19,659 (10.4%)	**<0.0001**
No	41,620	73,891 (9.9)	3483 (8.8)	
*4. Ophthalmic Factors (VA)*				
Good (≥20/66)	207,562	73,316 (65.9%)	134,246 (71.0%)	**0.0186**
Bad (<20/66)	6642	2253 (2.0%)	4389 (2.3%)	

*n*, numbers; DM, diabetes mellitus; HTN, hypertension; CHD, chronic heart disease; AMD, age-related macular degeneration; BMI, body mass index; Q, quartile; VA, visual acuity. Values are shown as mean ± standard deviation. Income grade: divided into quartiles; Q1: <25%, Q2: 25–50%, Q3: 50–75%, and Q4: >75–100%. BMI: computed by dividing an individual’s weight (kilograms) by the square of their height (meters). HTN: determined by either the prescription of anti-hypertensive drugs (≥1 per year) with ICD-10-CM codes (I10–I13, I15) or having a BP of more than 140/90 mmHg. Dyslipidemia: determined by a total cholesterol level ≥ 240 mg/dL or a prescription for medication with ICD-10-CM code E78. Heavy drinkers: determined as individuals who consumed ≥ 30 g of alcohol per day. ^a^ Missing BMI values: 4577 subjects in the no surgery group and 3016 subjects in the surgery group. ^b^ Missing smoking values: 56,120 subjects in the no surgery group and 29,935 subjects in the surgery group.

**Table 2 jcm-12-06441-t002:** Crude and adjusted HRs and 95% CIs for dementia risk.

Cataract Surgery	Subject No.	Case No.	Event (%)	Crude	Model 1 *	Model 2 **	Model 3 ***	Model 4 ****
				HR [95% CI]	HR [95% CI]	HR [95% CI]	HR [95% CI]	HR [95% CI]
No	189,119	18,879	10.0	1.00 (reference)	1.00 (reference)	1.00 (reference)	1.00 (reference)	1.00 (reference)
Yes	111,208	14,016	12.6	1.352 (1.322, 1.382)	1.155 (1.1, 1.181)	1.217 (1.181, 1.254)	1.243 (1.167, 1.323)	1.1 (1.021, 1.185)

HR, hazard ratio; CI, confidence interval. * adjusted for age and sex. ** adjusted for age, sex, and BMI. *** adjusted for age, sex, income, smoking history, drinking habits, and BMI. **** adjusted for age, sex, income VA, smoking history, drinking habits, BMI, and comorbidities (DM, HTN, dyslipidemia, stroke, depression, CHD, glaucoma, diabetic retinopathy, and AMD).

**Table 3 jcm-12-06441-t003:** Impact of cataract surgery on dementia risk via multivariate time-varying Cox regression analyses.

Subgroup and Variables				aHR (95% CI) ^†^	*p*-Value
Subgroups	Cataract Surgery	Visual Acuity	Subject Number		
**Whole population**	No	bad	4499	1 (ref.)	
	No	good	184,620	0.881 (0.714, 1.087)	0.236
	Yes	bad	2074	1.331 (0.967, 1.83)	0.0791
	Yes	good	109,134	0.879 (0.711, 1.085)	0.2292
**Age ≥ 65 years**	No	bad	3434	1 (ref.)	
	No	good	102,090	0.888 (0.71, 1.111)	0.3002
	Yes	bad	1588	1.372 (0.981, 1.917)	0.0642
	Yes	good	71,679	0.886 (0.708, 1.109)	0.2889
**Age < 65 years**	No	bad	1,065	1 (ref.)	
	No	good	82,530	0.903 (0.804, 1.013)	0.0805
	Yes	bad	486	1.057 (0.925, 1.209)	0.417
	Yes	good	37,455	**0.88 (0.785, 0.987)**	**0.0288**
**Male sex**	No	bad	1024	1 (ref.)	
	No	good	75,331	0.485 (0.264, 0.888)	0.0191
	Yes	bad	515	0.855 (0.295, 2.481)	0.7734
	Yes	good	42,204	**0.486 (0.263, 0.899)**	**0.0215**
**Female sex**	No	bad	3475	1 (ref.)	
	No	good	109,289	1.144 (0.827, 1.583)	0.4166
	Yes	bad	1559	1.456 (0.905, 2.343)	0.1218
	Yes	good	66,930	1.028 (0.742, 1.424)	0.8684

HR, hazard ratio; CI, confidence interval; Ref, reference. ^†^ Adjusted for age, sex, income, VA, smoking history, drinking habits, BMI, and comorbidities (DM, HTN, dyslipidemia, stroke, depression, CHD, glaucoma, diabetic retinopathy, and AMD).

## Data Availability

The datasets produced and/or examined during the course of the present study can be obtained from the corresponding authors upon reasonable request.

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
