# Peer review of "The Effect of Cataract Surgery on the Risk of Dementia: A Nationwide Cohort Study"

_jcm, 2023, doi:10.3390/jcm12206441_

Round 1

Reviewer 1 Report

The authors present a very interesting study about cataract surgery and risk of dementia in a nationwide population-based cohort population.

I congratulate the authors. I would just point our minor issues to be clarified:

  1. 1. Abstract. The authors state that ‘Advanced age and female sex are well-known risk factors for both cataract and dementia.’ Please give references where being a female is a risk factor for cataracts.

  1. 2. Abstract. Conclusion. The authors state that ‘Modern cataract surgery shows high efficacy and few safety concerns.’ This sentence is not a conclusion of this study because the authors did not study the safety of cataract surgery. Please place this sentence in the introduction of the manuscript and give references.

  1. 3. Introduction. The authors say that ‘low education level’ is a common factor for dementia and cataracts. Please give references of well-performed studies in which low education level is a risk factor for cataracts.

Author Response

3. Point-by-point response to Comments and Suggestions for Authors

Comments 1: The authors present a very interesting study about cataract surgery and risk of dementia in a nationwide population-based cohort population.

I congratulate the authors. I would just point our minor issues to be clarified:

Response 1: Thank you very much for taking time out of your busy schedule to review this article and for the valuable opinion.

Comments 2: Abstract. The authors state that ‘Advanced age and female sex are well-known risk factors for both cataract and dementia.’ Please give references where being a female is a risk factor for cataracts.

Response 2: Thank you for pointing out this. We totally agree with your opinion. And we cited the research, “Jefferis JM, Mosimann UP, Clarke MP. Cataract and cognitive impairment: a review of the literature. Br J Ophthalmol. 2011;95(1):17-23”. Therefore, we added references as follows.

<In the revised manuscript, 2nd page , “Abstract”, 1st line>

Advanced age and female sex are widely recognized risk factors for both cataract and dementia 1.

Comments 3: Abstract. Conclusion. The authors state that ‘Modern cataract surgery shows high efficacy and few safety concerns.’ This sentence is not a conclusion of this study because the authors did not study the safety of cataract surgery. Please place this sentence in the introduction of the manuscript and give references.

Response 3: Thank you for pointing out this. We totally agree with your opinion that our study did not evaluate the efficacy and safety of modern cataract surgery. Therefore, we deleted the sentence and we thought that it would be better not to mention in our manuscript.

Comments 4: Introduction. The authors say that ‘low education level’ is a common factor for dementia and cataracts. Please give references of well-performed studies in which low education level is a risk factor for cataracts.

Response 4: Thank you for pointing out this. We totally agree with your opinion. And we cited the research, “Jefferis JM, Mosimann UP, Clarke MP. Cataract and cognitive impairment: a review of the literature. Br J Ophthalmol. 2011;95(1):17-23”. Therefore, we added references as follows.

<In the revised manuscript, 4th page , “Introduction”, 1st paragraph, 6th line>

This relationship might be due to their sharing of common risk factors, particularly old age1,6, hypertension (HTN)1,6, diabetes mellitus (DM) 2,6 and low education level1.

4. Response to Comments on the Quality of English Language

Point 1: English language fine. No issues detected

Response : Thank you very much for taking time out of your busy schedule to review this article and for the valuable opinion.

Reviewer 2 Report

Very good study. In addition to the statistical analysis of the data: the description of the statistical methods does not indicate what the +- means, whether the number after the +- is the standard error or the standard deviation; it appears in the following results section, but it is good practice to indicate it already in the description of statistical methods.

And I doubt if all the quantitative features corresponded to a normal distribution, so in many places it was necessary to use the median and IQR, and also to indicate what statistical test was used to test the normal distribution.

And I have some specific questions:

What is the optimal time to have cataract surgery to reduce the risk of dementia?

How does the type of cataract surgery affect the risk of dementia?

Author Response

3. Point-by-point response to Comments and Suggestions for Authors

Comments 1: Very good study. In addition to the statistical analysis of the data: the description of the statistical methods does not indicate what the +- means, whether the number after the +- is the standard error or the standard deviation; it appears in the following results section, but it is good practice to indicate it already in the description of statistical methods.

Response 1: Thank you very much for taking time out of your busy schedule to review this article and for the valuable opinion. We mentioned that “Continuous variables are presented as mean ± standard deviation (SD) values” in the Methods section. We agree that it can cause confusion, so we added “Values are shown as mean ± standard deviation.” in the Table 1, too.

<In the revised manuscript, 8th page , “Method and Material” “2.5. Statistical Procedures”, 1st paragraph, 1th line>

We performed Kolmogorov-Smirnov test on continuous variables and p-value were >0.05, which was consistent with normal distribution. Therefore, continuous variables were expressed as mean ± standard deviation (SD).

Comments 2: And I doubt if all the quantitative features corresponded to a normal distribution, so in many places it was necessary to use the median and IQR, and also to indicate what statistical test was used to test the normal distribution.

Response 2: Thank you for pointing out this. We totally agree with your opinion. We performed Kolmogorov-Smirnov test on continuous variables and all of the variables (age and follow-up period) showed normal distribution (Kolmogorov-Smirnov test, Age; p=0.2 & follow-up period; p=0.38) Therefore, we supplemented the statistical method and the result, as follows.

<In the revised manuscript, 8th page , “Method and Material” “2.5. Statistical Procedures”, 1st paragraph, 1st line>

2.5. Statistical Procedures

We performed Kolmogorov-Smirnov test on continuous variables and p-value were >0.05, which was consistent with normal distribution. Therefore, continuous variables were expressed as mean ± standard deviation (SD).

<In the revised manuscript, 9th page, “Result” “3.1. Initial Characteristics of the Study Cohort”, 1st paragraph, 1th line>

The age and follow-up periods in the surgery and non-surgery groups were normally distributed (Kolmogorov-Smirnov test, Age; p=0.2 & follow-up period; p=0.38).

Comments 3: And I have some specific questions:

What is the optimal time to have cataract surgery to reduce the risk of dementia?

Response 3:

Thank you for pointing out this. While there is a common consensus that cataract surgery can reduce the dementia risk, the optimal timing of cataract surgery to reduce the risk of dementia remain areas of active investigation.

While cataract surgery is generally considered a safe and effective procedure, there can be some disadvantages to having it done so earlier, especially when the cataracts are not significantly affecting your vision or quality of life. Some potential disadvantages of early cataract surgery include unnecessary cost, surgical risks and recovery time, potential for miscorrection and decreased depth of focus.

 Therefore, it’s essential to have a thorough consideration with patients to weigh the pros and cons of cataract surgery based on each patient’s situation. And it is very important to consider patient’s subjective discomfort due to cataract. Nevertheless, our study suggests that in order to alleviate dementia risk, patients with potentially good vision after surgery, such as those with a younger age and relatively good VA and high risk for dementia, can be candidate for early surgery.

 Thank you for your precious opinion, and we regarded following sentence is consistent with author’s opinion about appropriate timing for surgery.

<In the revised manuscript, 19th page, “Discussion”, 3rd paragraph, 2nd line>

Cataract surgery should be recommended at an appropriate time, especially in patients with potentially good vision after surgery, such as those with a younger age and relatively good VA, as it can alleviate the risk of dementia.

Comments 4: How does the type of cataract surgery affect the risk of dementia?

Response 2: There is ongoing research into the potential relationship between cataract surgery and the risk of dementia, but the type of cataract surgery itself has not been definitively linked to dementia risk. The primary focus of cataract surgery is to improve vision and quality of life in individuals with cataracts. Any potential impact on dementia risk is still a subject of investigation and has not been conclusively established.

It's important to note that cataract surgery can involve different techniques and types of intraocular lenses (IOLs), such as monofocal, multifocal, or toric lenses. These variations in surgical approach and IOL choice are primarily aimed at addressing specific visual needs and correcting refractive errors.

While cataract surgery can improve vision and, in some cases, may have indirect benefits on cognitive function (by enabling individuals to remain socially and mentally active due to improved vision), the relationship between the type of cataract surgery and dementia risk is not a well-established factor in the decision-making process for cataract surgery. According to et al., who involved 25 patients with multifocal surgery group and 26 patients with monofocal surgery group, IOL type did not show significant difference on cognitive dysfunction.

Thank you for your previous comments and we supplemented this relationship between cataract surgery type and dementia risk in the “Discussion”.

<In the revised manuscript, 18th page, “Discussion” 4th paragraph, 1th line>

Cataract surgery involves various types of intraocular lenses (IOLs), such as monofocal, multifocal, or toric lenses 22. These variations in surgical approach and IOL choice are primarily aimed at addressing specific visual needs and correcting refractive errors 22. However, the relationship between the type of cataract surgery and dementia risk is not a well-established factor in the decision-making process for cataract surgery 22,23.   According to et al., who involved 25 patients with multifocal surgery group and 26 patients with monofocal surgery group, IOL type did not show significant difference on cognitive dysfunction 23. However, enrolled patient was small and covariates were not adjusted 23.  Further research considering this relationship is required, with large subjects.

4. Response to Comments on the Quality of English Language

Point 1: I am not qualified to assess the quality of English in this paper

Response : Thank you very much for taking time out of your busy schedule to review this article and for the valuable opinion.
